# xLSTM-UNet can be an Effective Backbone for 2D & 3D Biomedical Image Segmentation Better than its Mamba Counterparts

1st Tianrun Chen[+]
*College of Computer Science and Technology*
*Zhejiang University*
Hangzhou, China
tianrun.chen@zju.edu.cn

1st Chaotao Ding[+]
*KOKONI, Moxin (Huzhou) Technology Co., LTD.*
Huzhou, China
2021388117@stu.zjhu.edu.cn

1st Lanyun Zhu[+]
*Singapore University of Technology and Design*
Singapore, 487372
lanyun_zhu@mymail.sutd.edu.sg

1st Tao Xu[+]
*Huzhou University*
Huzhou, China
yhhlllaini@gmail.com

5th Yan Wang
*Beihang University*
Beijing, China
wangyan9509@gmail.com

6th Deyi Ji
*University of Science and Technology of China*
Hefei, China
jideyi16@foxmail.com

7th Ying Zang[*]
*School of Information Engineering*
*Huzhou University*
Huzhou, China
02750@zjhu.edu.cn

8th Zejian Li
*School of Software Technology*
*Zhejiang University*
Hangzhou, China
zejianlee@zju.edu.cn

*Abstract*—In this work, we replace Mamba in UMamba with recent xLSTM, and surprisingly, it works well! Convolutional Neural Networks (CNNs) and Vision Transformers (ViT) have been pivotal in biomedical image segmentation. Yet, their ability to manage long-range dependencies remains constrained by inherent locality and computational overhead. To overcome these challenges, in this technical report, we first propose xLSTM-UNet, a UNet structured deep learning neural network that leverages Vision-LSTM (xLSTM) as its backbone for medical image segmentation. xLSTM has recently been proposed as the successor of Long Short-Term Memory (LSTM) networks and has demonstrated superior performance compared to Transformers and State Space Models (SSMs) like Mamba in Neural Language Processing (NLP) and image classification (as demonstrated in Vision-LSTM, or ViL implementation). Here, we provide the first integration of xLSTM with image segmentation backbone – namely xLSTM-U, which extend the success of xLSTM in the biomedical image segmentation domain. By integrating the local feature extraction strengths of convolutional layers with the long-range dependency-capturing abilities of xLSTM, the proposed xLSTM-UNet offers a robust solution for comprehensive image analysis. We validate the efficacy of xLSTM-UNet through experiments. Our findings demonstrate that xLSTM-UNet consistently surpasses the performance of leading CNN-based, Transformer-based, and Mamba-based segmentation networks in multiple datasets in biomedical segmentation including organs in abdomen MRI, instruments in endoscopic images, and cells in microscopic images. With comprehensive experiments performed, this paper highlights the potential of xLSTM-based architectures in advancing biomedical image analysis in both 2D and 3D. We believe this new finding will be of interest to the research community and may inspire future studies. The code, models, and datasets are publicly available at *https://github.com/tianrun-chen/xLSTM-UNet-PyTorch/tree/main*.

*Index Terms*—3D Medical Image Segmentation, Long Range Sequential Modeling, Long Short-Term Memory (LSTM), State Space Models, UNet, Vision Mamba, Vision Transformer, xLSTM

## I. INTRODUCTION

Biomedical image segmentation is a critical task in medical imaging, enabling precise delineation of anatomical structures and anomalies essential for diagnosis, treatment planning, and research [1], [2]. In recent years, deep learning methods have achieved remarkable success in tumor segmentation [3] and organ segmentation in 3D Computed Tomography (CT) scans [4], as well as in cell segmentation in microscopy images [5], [6]. These advancements underscore the transformative impact of deep learning on the landscape of biomedical image segmentation, paving the way for more accurate and efficient diagnostic and treatment planning tools. Traditionally, Convolutional Neural Networks (CNNs) have been the backbone of this domain in deep learning-enabled methods, leveraging their powerful local feature extraction capabilities [7]–[9]. More recently, Vision Transformers (ViTs) have gained popularity by offering a robust alternative, capable of capturing global context through self-attention mechanisms [10]–[12]. Despite their successes, both CNNs and ViTs face inherent limitations. CNNs struggle with long-range dependencies due to

+ Equal Contribution * Corresponding Author

their localized receptive fields, while ViTs encounter substantial computational overhead [13], [14], especially with high-resolution images or high-dimensional imaging modalities like 3D images or hyper-spectral imaging like stimulated Raman scattering (SRS) imaging [15]–[17] or mid-infrared (IR) spectroscopic imaging [18].

To address these challenges, recent work has proposed to integrate computation modules that have long-range dependencies and also exhibit linear computational and memory complexity w.r.t. sequence length. Among these computation modules, State Space Models (SSMs) [19]–[21], like Mamba [22], has demonstrated its huge success. SSMs excel in handling long-range dependencies and have been successfully integrated into conventional UNet architectures. Variants like UMamba [23], VM-Unet [24]–[26], Mamba-Unet [27], Swin-UMamba [28], and SegMamba [29], have demonstrated their considerable success.

Meanwhile, Extended Long-Short Term Memory (xLSTM) has recently emerged as a powerful successor to Long Short-Term Memory (LSTM) networks, challenging Transformers in sequence modeling [30]. Like SSMs, xLSTM can handle long-range dependencies and maintain linear computational and memory complexity. However, xLSTM has demonstrated superior performance in neural language processing (NLP) and image classification (in its Vision-LSTM (ViL) implementation [31]). This success naturally raises the question: **Can xLSTM, or ViL, also excel in image segmentation, specifically in the field of medical image segmentation?**

The answer is *Yes!* In this technical report, we introduce xLSTM-UNet, the first xLSTM-enabled U-Net image segmentation network that can perform both 2D and 3D medical image segmentation tasks and achieves state-of-the-art (SOTA) results. We conducted comprehensive experiments in various 2D and 3D medical segmentation scenarios, including organs in abdominal MRI, instruments in endoscopy, cells in microscopy, and cancer segmentation in 3D brain MRI volumes. The results show that xLSTM-UNet outperforms existing CNN-based and Transformer-based segmentation methods, as well as its Mamba-based counterparts. These findings highlight the potential of xLSTM-based architecture to set new benchmarks in the field of medical image segmentation, offering improved accuracy and efficiency across a wide range of applications. To further advance research in this area, we will release the model and code, enabling future explorations in various fields such as automated pathology detection, camouflaged image segmentation, precision agriculture, environmental monitoring, satellite imagery analysis, and industrial inspection, and so on.

## II. RELATED WORK

Medical image segmentation is a crucial application of computer vision, involving the delineation of pathological and anatomical structures from volumetric data for diagnosis, treatment planning, and patient monitoring. This task is complex due to the multidimensional nature of the data, requiring highly accurate segmentation outputs.

Deep learning has significantly advanced medical image segmentation, particularly with convolutional neural networks (CNNs) [32]. Early works [8], [33] used CNNs as a backbone for segmentation tasks. A key breakthrough came with U-Net [7], which introduced a symmetric encoder-decoder structure that captured both fine-grained spatial information and abstract features. Extensions like Unet++ [9] improved performance through nested skip connections and deep supervision. However, CNNs often struggle with long-range dependencies.

Transformers [34] addressed this limitation by modeling global dependencies with self-attention mechanisms, as seen in hybrid models like TransUNet [10], which combine CNNs for local features and Transformers for global context.

While Transformers improve segmentation, they are computationally expensive. To mitigate this, State Space Models (SSMs) [19]–[21], such as Mamba [22], offer efficient long-range dependency handling with linear computational complexity, as demonstrated in Unet-based models [23]–[29].

Extended Long-Short Term Memory (xLSTM) networks [30] present another alternative, efficiently modeling long-range dependencies with reduced computational demands. Vision-xLSTM [31] integrates global context with computational efficiency, making it a promising approach for developing precise and efficient medical image segmentation models.

## III. METHOD

*1) Vision-xLSTM:* Vision-xLSTM (ViL) is an adaptation of xLSTM specifically for computer vision tasks. The configuration contains interchanging mLSTM segments where even-indexed segments handle flattened features starting from the top left moving to the bottom right, while odd-indexed segments proceed from the bottom right to the top left. This two-way operation allows ViL segments to grasp strong worldwide dependencies within the input.

The flattened features, after being normalized, are projected into an embedding domain, doubling their size. These enlarged embeddings are split into two paths: $x_{mlstm} \in \mathbb{R}^{N \times 2Z}$ and $y \in \mathbb{R}^{N \times 2Z}$. $x_{mlstm}$ undergoes a 1D causal convolution process with the SiLU activation function applied to it. The intermediate outcome ($X \in \mathbb{R}^{N \times 2Z}$) is subsequently mapped to query, key, and value vectors, which are akin to the vectors used in the Transformer model. These vectors are then forwarded to the mLSTM cell. The mLSTM sublayer consists of n parallel attention heads, each equipped with a matrix memory unit.

At first, input and forget gate pre-activations, $\tilde{i} \in \mathbb{R}^{N \times d_h}$ and $\tilde{f} \in \mathbb{R}^{N \times d_h}$ respectively, are calculated by linearly projecting the concatenated $Q$, $K$, and $V$ matrices as expressed below mathematically:

$$\tilde{i} = W_I[Q, K, V] + B \tag{1}$$

$$\tilde{f} = W_F[Q, K, V] + B \tag{2}$$

Here, $B \in \mathbb{R}^{N \times d_h}$ is the bias matrix. The formation of query, value, and key is mathematically defined as:

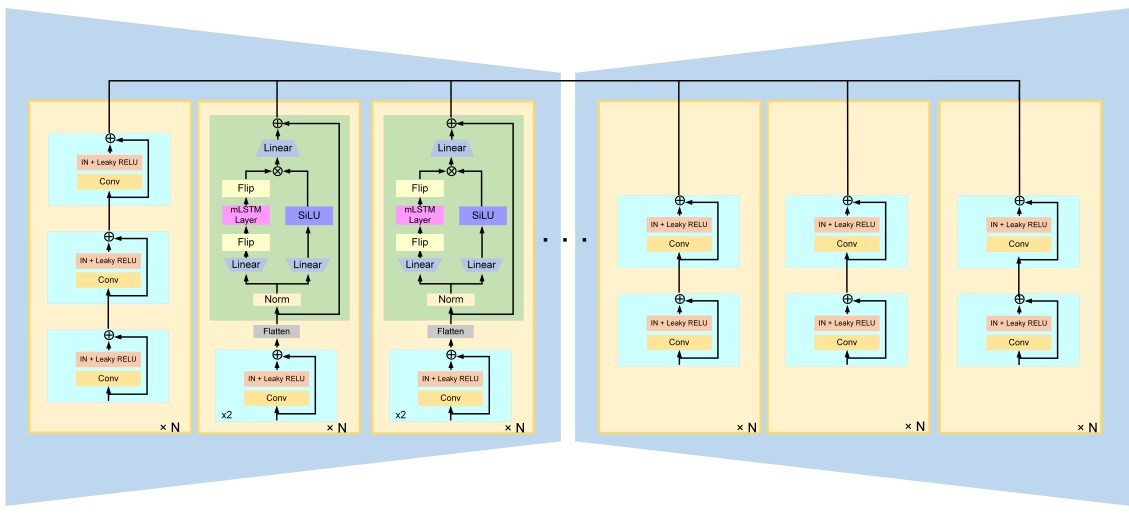

Fig. 1. The framework of the proposed xLSTM-UNet, which is based on the conventional U-Net structure. xLSTM is used in multi-resolution feature extraction process in the encoder.

$$Q = XW_Q^T, K = XW_K^T, V = XW_V^T \tag{3}$$

Where $Q$ belongs to $\mathbb{R}^{N \times d}$, $K$ resides in $\mathbb{R}^{N \times d}$, $V$ is in $\mathbb{R}^{N \times d}$, these are query, key, and value matrices. The $W_Q$ resides in $\mathbb{R}^{2Z \times d}$, $W_K$ is within $\mathbb{R}^{2Z \times d}$, and $W_V$ is found in $\mathbb{R}^{2Z \times d}$, they are the adjustable weight matrices for generating the query, key, and value per head. Herein, $d$ symbolizes the intended dimension for queries, keys, and values.

The pre-activation outputs further derive the gate decay matrix $D \in \mathbb{R}^{N \times d_h}$ following this equation:

$$D = \tilde{i} \oplus \log \sigma(\tilde{f}) \tag{4}$$

Herein, $\log \sigma$ denotes the log-sigmoid activation function. The decay matrix $D$ is stabilized to ensure subsequent exponentiation of $D$ results in stable output. The final gate decay matrix output $\hat{D} \in \mathbb{R}^{N \times d_h}$ is represented as:

$$\hat{D} = \exp(D) \tag{5}$$

Here, $\exp$ represents the exponentiation operation. The $Q$ and $K$ vectors perform a dot product to derive the attention scores $S \in \mathbb{R}^{N \times N}$, analogous to the self-attention mechanism in Transformers. A causal mask $M \in \mathbb{R}^{N \times N}$ ensures that attention is solely from the previous patch to the current one. It is mathematically represented as:

$$S = \text{Softmax}(\frac{QK^\top}{\sqrt{d}} + M) \tag{6}$$

The attention score matrix undergoes element-wise multiplication with $\hat{D}$ to form the combination matrix $C \in \mathbb{R}^{N \times d_h}$.

$$C = S^\top \hat{D} \tag{7}$$

Finally, the cell state $\tilde{h}_t$ is updated as:

$$\tilde{h}_t = C \otimes V \tag{8}$$

Then, the embedding $y$ from the second path is multiplied with $\tilde{h}_t$, followed by a down-projection operation. This final output $o$ is projected into a $Z$-dimensional embedding space using the projection matrix $W_D \in \mathbb{R}^{2Z \times Z}$. Mathematically, this can be expressed as:

$$o = W_D^\top(y \otimes \tilde{h}_t) \tag{9}$$

*2) The Architecture of xLSTM-UNet:* Fig. 1 showcases the xLSTM-UNet network architecture. xLSTM-UNet follows a conventional UNet-like structure. The input information first passes a convolution layer for initial down-sampling. Then several subsequent layers that are constructed using the aforementioned xLSTM building blocks to capture both local features and long-range dependencies form the main part of the encoder. Note that the xLSTM-UNet is designed with the goal of harnessing the best aspects of both U-Net and xLSTM for improved global comprehension in medical image understanding. Therefore, instead of only applying xLSTM in the compressed latent space after the down-sampling has finished, we hereby use the xLSTM in multiple layers in the encoder, in which each layer contains two successive Residual blocks with one plain convolution and an Instance Normalization (IN) and followed by a xLSTM block as in [31]. Specifically, the image features passing the residual blocks has a shape of $(B, C, H, W, D)$, which is first flattened and transposed to $(B, H \times W \times D, C)$, followed by a layer normalization, and then feed to the ViL block. Such the practice of involving xLSTM in multiple layers helps the feature extractions in multiple resolutions/perception fields, and this information extracted by xLSTM blocks is reshaped to $(B, C, H, W, D)$ and concatenated to the layers in the decoding steps to facilitate the segmentation mask generation.

After encoding, the decoder, comprising Residual blocks and transposed convolutions, concentrates on the meticulous recovery of detailed local information. Additionally, we inherit

the skip connection from the U-Net architecture to interconnect the hierarchical features from the encoder to the decoder. The final decoder feature is fed into a 1 x 1 convolutional layer, coupled with a Softmax layer, to generate the ultimate segmentation probability map. Furthermore, following [23] in 2D segmentation, we also implemented a variant where the U-xLSTM block is exclusively utilized in the bottleneck, denoted as 'ours_bot', while 'ours_enc,' denotes the network that applies xLSTM block in all encoder blocks in all 2D segmentation tasks. In 3D segmentation, the xLSTM blocks is added in the bottleneck.

## IV. EXPERIMENTS

### A. Datasets

To validate the effectiveness of our xLSTM-UNet, we utilized several representative medical image segmentation datasets, covering organ, instrument, and cell segmentation across different resolutions and modalities. This comprehensive evaluation demonstrates the applicability and superiority of xLSTM-UNet across diverse medical imaging scenarios.

**Abdomen MRI:** We used the Abdomen MRI dataset from the MICCAI 2022 AMOS Challenge [4] for abdominal organ segmentation. The dataset, annotated by radiologists using MedSAM [2] and ITK-SNAP [35], includes 60 MRI scans for training and 50 for testing, with 13 organ classes. For 2D tasks, images were cropped to 320x320 pixels, and for 3D tasks, patches were 48x160x224 pixels, following U-Mamba settings [23].

**Endoscopy images:** The endoscopy image dataset was sourced from the MICCAI 2017 EndoVis Challenge [36], focusing on the segmentation of seven surgical instruments from endoscopic images. We used the official split of 1800 training and 1200 testing frames, with images resized to 384x640 pixels for nnU-Net.

**Microscopy images:** The microscopy image dataset was obtained from the NeurIPS 2022 Cell Segmentation Challenge [6], which focuses on cell segmentation in various microscopy images. We trained on 1000 images and tested on 101, converting the task to semantic segmentation as in [23], with images resized to 512x512 pixels for nnU-Net.

**BraTS2023:** The BraTS2023 dataset [37]–[39] comprises 1,251 3D brain MRI volumes. Each volume features four imaging modalities (T1, T1Gd, T2, and T2-FLAIR) and three segmentation targets: Whole Tumor (WT), Enhancing Tumor (ET), and Tumor Core (TC). For training, we use a random crop size of 128×128×128 to process the 3D data.

### B. Implemetation details

The network is implemented based on UMamba [23]. The loss function used is the sum of Dice loss and cross-entropy loss. We employ the AdamW optimizer with a weight decay of 0.05. The learning rates were set to 0.005 for the Abdomen MRI dataset, 0.01 for the Endoscopy dataset, 0.007 for training xLSTM-UNet_Bot on the Microscopy dataset, 0.0015 for training xLSTM-UNet_Enc on the Microscopy dataset, and 0.01 for training on the BraTS2023 dataset. The batch sizes

were set as follows: 2 for the 3D Abdomen MRI dataset, 30 for the 2D Abdomen MRI dataset, 2 for the Endoscopy dataset, 12 for the Microscopy dataset, and 4 for the BraTS2023 dataset. All networks were trained from scratch for 1000 epochs on a single NVIDIA A100 GPU. For more implementation details, please refer to our codebase.

### C. Baselines

In 2D medical segmentation, to ensure a fair comparison, we follow the evaluation protocol in UMamba [23]. We selected two CNN-based segmentation networks (nnU-Net [4] and SegResNet [40]) and two Transformer-based networks (UNETR [12] and SwinUNETR [11]), as well as the UMamba itself, which has two variations: U-Mamba_Bot and U-Mamba_Enc. Similar to our configuration, U-Mamba_Bot is applied only at the bottleneck, while U-Mamba_Enc is used in each encoder. We used the Dice Similarity Coefficient (DSC) and Normalized Surface Distance (NSD) as evaluation metrics for the semantic segmentation tasks on the Abdomen MRI and Endoscopy datasets [41]. For cell segmentation on the Microscopy dataset, we employed the F1 score.

In 3D medical segmentation, for the 3D Abdomen MRI dataset, the baseline methods and tasks remain consistent with those used in 2D medical segmentation. For the BraTS2023 dataset, to ensure a fair comparison, we follow the evaluation protocol outlined in SegMamba [29]. We use the same baseline methods, including three CNN-based methods (SegresNet [40], UX-Net [42], MedNeXt [43]), three transformer-based methods (UNETR [12], SwinUNETR [11], and SwinUNETR V2 [44]), and the Mamba-based method SegMamba itself. Following previous evaluation protocols, Dice and HD95 were used as evaluation metrics.

### D. Quantitative and Qualitative Results for 2D Segmentation

Table I presents the segmentation performance of various methods on the Abdomen MRI 2D, Endoscopy, and Microscopy datasets for 2D image segmentation task. Our proposed xLSTM-UNet outperforms all baseline methods and achieves state-of-the-art (SOTA).

Notably, both variations of xLSTM-UNet show superior performance across all datasets. Specifically, xLSTM-UNet_Enc demonstrates the highest performance with a DSC of 0.7747 and an NSD of 0.8374 on the Abdomen MRI 2D dataset, outperforming the previous state-of-the-art (SOTA) model, U-Mamba by a significant margin. Additionally, xLSTM-UNet_Bot achieves DSC and NSD scores of 0.7636 and 0.8322, respectively, surpassing the similarly structured U-Mamba_Bot. Similarly, on the Endoscopy dataset, both xLSTM-UNet_Bot and xLSTM-UNet_Enc achieve the best DSC and NSD scores of 0.6843 and 0.7001, respectively. For the Microscopy dataset, xLSTM-UNet_Enc and xLSTM-UNet_Bot achieve F1 scores of 0.6036 and 0.5818, respectively, both surpassing the previous SOTA results, indicating their robustness in cell segmentation tasks.

The visualized segmentation examples of 2D medical images further illustrate the effectiveness of xLSTM-UNet. As

TABLE I
PERFORMANCE COMPARISON OF DIFFERENT METHODS

| Methods | Organs in Abdomen MRI 2D | | Instruments in Endoscopy | | Cells in Microscopy |
| | DSC ↑ | NSD ↑ | DSC ↑ | NSD ↑ | F1 ↑ |
|---|---|---|---|---|---|
| nnU-Net | 0.7450 ± 0.1117 | 0.8153 ± 0.1145 | 0.6264 ± 0.3024 | 0.6412 ± 0.3074 | 0.5383 ± 0.2657 |
| SegResNet | 0.7317 ± 0.1379 | 0.8034 ± 0.1386 | 0.5820 ± 0.3268 | 0.5968 ± 0.3303 | 0.5411 ± 0.2633 |
| UNETR | 0.5747 ± 0.1672 | 0.6309 ± 0.1858 | 0.5017 ± 0.3201 | 0.5168 ± 0.3235 | 0.4357 ± 0.2572 |
| SwinUNETR | 0.7028 ± 0.1348 | 0.7669 ± 0.1442 | 0.5528 ± 0.3089 | 0.5683 ± 0.3123 | 0.3967 ± 0.2621 |
| U-Mamba_Bot | 0.7588 ± 0.1051 | 0.8285 ± 0.1074 | 0.6540 ± 0.3008 | 0.6692 ± 0.3050 | 0.5389 ± 0.2817 |
| U-Mamba_Enc | 0.7625 ± 0.1082 | 0.8327 ± 0.1087 | 0.6303 ± 0.3067 | 0.6451 ± 0.3104 | 0.5607 ± 0.2784 |
| Ours_bot | 0.7636 ± 0.1006 | 0.8322 ± 0.1034 | **0.6843 ± 0.3005** | **0.7001 ± 0.3046** | 0.5818 ± 0.2386 |
| Ours_enc | **0.7747 ± 0.0950** | **0.8374 ± 0.0951** | **0.6843 ± 0.3024** | **0.7001 ± 0.3067** | **0.6036 ± 0.2435** |

shown in Figure 2, xLSTM-UNet is more robust to heterogeneous appearances and exhibits fewer segmentation outliers compared to other models. This visual evidence underscores the quantitative results, highlighting the superior performance and reliability of xLSTM-UNet in diverse medical image segmentation tasks.

*E. Quantitative and Qualitative Results for 3D Segmentation*

3D medical image segmentation is generally more challenging compared to its 2D counterpart, as it involves processing a larger amount of information. The increased dimensionality leads to a dramatic surge in computational complexity, with the resolution increase causing a cubic rise in the number of computations. Accurate spatial relationship modeling is also essential for achieving satisfactory segmentation results. These factors make the xLSTM-based building blocks, with their computational efficiency, well-suited for this task.

We conducted evaluations on the 3D segmentation dataset in BraTS2023 and Abdomen MRI 3D. Table II shows the performance comparison on the BraTS2023 dataset, including metrics for whole tumor (WT), tumor core (TC), and enhancing tumor (ET) regions. Our proposed method demonstrates superior performance across all evaluated metrics, including Dice and HD95, compared to other baseline methods such as SegresNet, UX-Net, MedNeXt, UNETR, SwinUNETR, SwinUNETR-V2, and SegMamba. Specifically, our method achieves the highest average Dice score of 91.80, highlighting its effectiveness in accurately segmenting brain tumor regions. Table III shows the performance comparison of the Organs in the Abdomen MRI 3D dataset. Our proposed method, xLSTM-UNet_Bot, achieves the highest Dice score of 0.8483 and the best NSD score of 0.9153, surpassing other methods such as nnU-Net, SegResNet, UNETR, SwinUNETR, and U-Mamba_Bot. This demonstrates the robustness and accuracy of our approach in segmenting abdominal organs in MRI images.

The ability of xLSTM to effectively model semantic information in complex spatial domains has been a key factor in our success. The superior experimental results also underscore the suitability of xLSTM-based building blocks for tackling the challenge of semantic segmentation in complex imaging applications.

*F. Ablation Studies*

To further validate the improvements introduced by our newly proposed modules, we conducted a series of ablation

TABLE II
PERFORMANCE COMPARISON ON BRATS2023 DATASET FOR 3D IMAGE SEGMENTATION

| Methods | WT | | TC | | ET | | Avg | |
| | Dice↑ | HD95↓ | Dice↑ | HD95↓ | Dice↑ | HD95↓ | Dice↑ | HD95↓ |
|---|---|---|---|---|---|---|---|---|
| SegresNet | 92.02 | 4.07 | 89.10 | 4.08 | 83.66 | 3.88 | 88.26 | 4.01 |
| UX-Net | 93.13 | 4.56 | 90.03 | 5.68 | 85.91 | 4.19 | 89.69 | 4.81 |
| MedNeXt | 92.41 | 4.98 | 87.75 | 4.67 | 83.96 | 4.51 | 88.04 | 4.72 |
| UNETR | 92.19 | 6.17 | 86.39 | 5.29 | 84.48 | 5.03 | 87.68 | 5.49 |
| SwinUNETR | 92.71 | 5.22 | 87.79 | 4.42 | 84.21 | 4.48 | 88.23 | 4.70 |
| SwinUNETR-V2 | 93.35 | 5.01 | 89.65 | 4.41 | 85.17 | 4.41 | 89.39 | 4.51 |
| SegMamba | 93.61 | **3.37** | 92.65 | **3.85** | 87.71 | **3.48** | 91.32 | **3.56** |
| Ours | **93.84** | 3.89 | **92.77** | 4.06 | **88.79** | 4.14 | **91.80** | 4.03 |

TABLE III
PERFORMANCE COMPARISON ON ORGANS IN ABDOMEN MRI 3D DATASET

| Methods | DSC ↑ | NSD ↑ |
|---|---|---|
| nnU-Net | 0.8309 ± 0.0769 | 0.8996 ± 0.0729 |
| SegResNet | 0.8146 ± 0.0959 | 0.8841 ± 0.0917 |
| UNETR | 0.6867 ± 0.1488 | 0.7440 ± 0.1627 |
| SwinUNETR | 0.7565 ± 0.1394 | 0.8218 ± 0.1409 |
| U-Mamba | 0.8453 ± 0.0673 | 0.9121 ± 0.0634 |
| Ours | **0.8483 ± 0.0774** | **0.9153± 0.0596** |

studies. The process began by using a baseline model where all blocks in the U-Net architecture were fully convolutional layers. Starting from this baseline, we incrementally replaced the convolutional blocks with xLSTM modules. The results of these ablation studies are presented in Table IV, illustrating the effectiveness of each module on overall model performance.

In our ablation experiments, replacing even a single convolutional layer with an xLSTM module resulted in significant improvements, particularly on the Dataset704_Endovis17 dataset, where DCS and NSD metrics increased by 18.9% and 19.1%, respectively. As more xLSTM layers were added, accuracy consistently improved across all datasets. These results demonstrate that integrating xLSTM blocks into the U-Net architecture effectively enhances segmentation accuracy in medical imaging.

## V. EFFICIENCY ANALYSIS

The efficiency analysis of different methods, as shown in Table V, highlights the performance and computational complexity of models. The scatter plot (Figure 3) visualizes the relationship between FLOPs and F1 scores for various models for the task of cell segmentation in microscopy image, with the circle size representing the number of parameters. Our models, Ours_bot and Ours_enc, demonstrate a significant

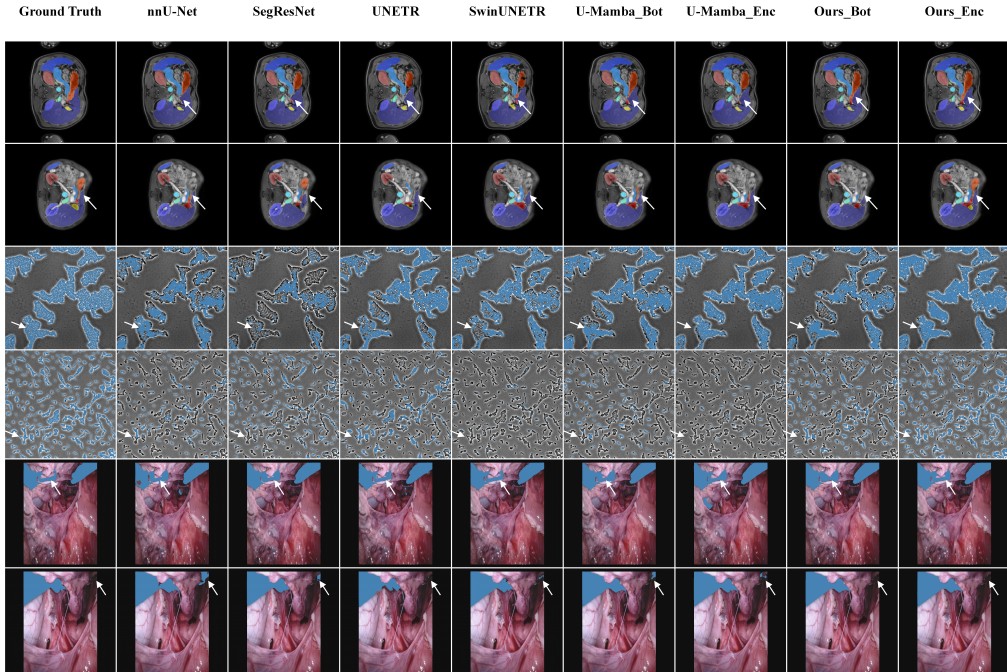

Fig. 2. Visualized examples of 2D medical segmentation in various dataset. xLSTM-UNet demonstrates greater robustness to heterogeneous appearances and exhibits fewer segmentation errors.

TABLE IV
PERFORMANCE COMPARISON OF DIFFERENT METHODS ON VARIOUS DATASETS

| Methods | Organs in Abdomen MRI 3D | | Cells in Microscopy | Instruments in Endoscopy | | Organs in Abdomen MRI 2D | | |
|---|---|---|---|---|---|---|---|---|
| | DSC ↑ | NSD ↑ | F1 ↑ | DSC ↑ | NSD ↑ | DSC ↑ | NSD ↑ | Inference Speed(s) |
| FCN | 0.8315 ± 0.1544 | 0.8961 ± 0.1467 | 0.5685 ± 0.2351 | 0.4737 ± 0.2968 | 0.4878 ± 0.3008 | 0.7402 ± 0.1955 | 0.8076 ± 0.1938 | 2.1051 |
| one_xlstm | 0.8459 ± 0.1299 | 0.9147 ± 0.1225 | 0.5835 ± 0.2453 | 0.6621 ± 0.3030 | 0.6786 ± 0.3074 | 0.7562 ± 0.1847 | 0.8222 ± 0.1868 | 6.0244 |
| two_xlstms | 0.8452 ± 0.1307 | 0.9123 ± 0.1226 | 0.5860 ± 0.2423 | 0.6683 ± 0.3026 | 0.6859 ± 0.3068 | 0.7692 ± 0.1755 | 0.8373 ± 0.1783 | 6.1156 |
| Ours_enc(three_xlstms) | **0.8483 ± 0.0774** | **0.9153± 0.0596** | **0.6036 ± 0.2435** | **0.6843 ± 0.3024** | **0.7001 ± 0.3067** | **0.7747 ± 0.0950** | **0.8374 ± 0.0951** | 6.4247 |

advantage in terms of efficiency and performance. Specifically, Ours_enc achieves the highest F1 score of 0.6036 with a moderate computational cost of 125.9G FLOPs, which is considerably lower than the FLOPs of UNETR (120.1G FLOPs with an F1 score of 0.4357) and nnFormer (136.7G FLOPs with an F1 score of 0.5818). Furthermore, Ours_bot maintains a competitive F1 score of 0.5818 with only 101.7G FLOPs, which is significantly more efficient than nnFormer and other high-parameter models like U-Mamba_Bot and U-Mamba_Enc. This analysis underscores the superior balance achieved by our models between computational efficiency and segmentation accuracy, making them particularly suitable for practical applications where both performance and resource consumption are critical. Our models' lower parameter counts and FLOPs, combined with high F1 scores, clearly demonstrate their superiority over existing methods, affirming the effectiveness of our approach in optimizing model architecture for both efficiency and accuracy.

We also evaluated the inference speed of the models as part of the ablation studies, focusing on the 2D segmentation task from the Dataset702_AbdomenMR dataset. The results, shown in the last column of Table IV, provide the average inference time per nii.gz file, highlighting the trade-off between performance and computation time. While the baseline FCN model had the fastest inference speed, the xLSTM-based models progressively increased inference time with each additional layer. However, the improvements in segmentation accuracy justify the added computation time for tasks requiring high precision.

## VI. DISCUSSION

This study demonstrates that xLSTM, a model with linear computational complexity, can be an effective component in image segmentation networks. Our experimental results clearly show that xLSTM-UNet outperforms Mamba-based counterparts, underscoring the promising future of xLSTM. Given the recent huge interest in Mamba in academia, we believe that it is important to also recognize and investigate the potential of xLSTM, which has shown remarkable efficacy in this domain.

Meanwhile, medical image segmentation is inherently challenging. General image segmentation foundation models, such as Segment Anything, often fail when applied to medical images [45]–[47]. Furthermore, medical imaging datasets are typically small. In this study, the datasets used were limited in size, which restricts our ability to explore the effects of

TABLE V
MODEL PARAMETERS AND FLOPS COMPARISON OF DIFFERENT METHODS IN DIFFERENT TASKS

| Methods | Organs in Abdomen MRI 2D | | Instruments in Endoscopy | | Cells in Microscopy | |
|---|---|---|---|---|---|---|
| | param number | FLOPs | param number | FLOPs | param number | FLOPs |
| nnU-Net | 33M | 23.3G | 33M | 55.9G | 46M | 60.1G |
| SegResNet | 6M | 24.5G | 6M | 58.9G | 6M | 62.8G |
| UNETR | 87M | 42.1G | 87M | 111.5G | 88M | 120.1G |
| SwinUNETR | 25M | 27.9G | 25M | 67.1G | 25M | 71.7G |
| nnFormer | 60M | 50.2G | 60M | 125.5G | 60M | 136.7G |
| U-Mamba_Bot | 63M | 45.7G | 63M | 109.7G | 86M | 117.8G |
| U-Mamba_Enc | 67M | 49.9G | 67M | 119.8G | 92M | 128.7G |
| Ours_bot | 42M | 41.2G | 47M | 99.2G | 64M | 101.7G |
| Ours_enc | 48M | 65.7G | 48M | 229.6G | 65M | 125.9G |

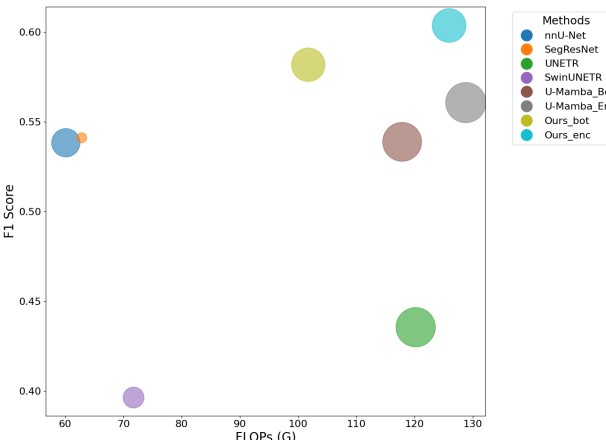

Fig. 3. The Visualization of Efficiency Analysis for Different Methods in Cells in Microscopy Segmentation. It is obvious that our approach achieves the highest F1 Score but with reasonable FLOPs and parameter numbers.

different network scales and dataset sizes on segmentation outcomes. Investigating whether xLSTM-driven image algorithms adhere to scaling laws remains an interesting question for future research.

Currently, xLSTM lacks dedicated optimization for hardware such as NVIDIA GPUs, which presents an opportunity for the community to contribute. Collaborative efforts are essential to optimize xLSTM for various vision tasks, leveraging its full potential. By releasing our code, we aim to encourage and facilitate further research and development, enabling the community to build on our initial findings and drive progress in this area.

This research represents an initial exploration into the application of xLSTM in medical image segmentation. There are many peaks to climb and numerous scenarios to test in this field. We hope that our comprehensive experiments and tests will demonstrate the significant potential of xLSTM in practical applications, encouraging scholars to continue exploring this promising model. With further development and optimization, we envision xLSTM achieving success comparable to that of Mamba and even Transformers, becoming a cornerstone in image segmentation and beyond.

## VII. CONCLUSION

In this paper, we introduce xLSTM-UNet, the first U-Net architecture enhanced with Extended Long-short-memory (xLSTM) / ViL for both 2D and 3D medical image segmentation tasks. Through extensive experiments across a variety of medical imaging scenarios—including abdominal MRI, endoscopy, microscopy, and brain MRI—we have demonstrated that xLSTM-UNet significantly outperforms existing CNN-based and Transformer-based methods, as well as its Mamba-based counterparts. These findings underscore the effectiveness of xLSTM in handling complex segmentation tasks, particularly in the challenging domain of 3D medical image segmentation.

Our results show that the xLSTM-based architecture can achieve state-of-the-art (SOTA) performance with reasonable memory and computation cost, offering enhanced accuracy and efficiency. This marks a significant advancement and future potential of xLSTM or similar building blocks, not only in the field of medical image segmentation, but also with potential applications extending beyond healthcare. We believe that this new findings will be interesting to the community and can inspire future researches.

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
