# OpenReview forum: "xLSTM-UNet can be an Effective Backbone for 2D & 3D Biomedical Image Segmentation Better than its Mamba Counterparts"
_IEEE.org/EMBS/BHI/2024/Conference — IEEE BHI'24_

### Official Review · Reviewer_65oJ · 2024-08-06
**An image segmentation model that combines xLSTM with U-Net that outperforms existing methods in biomedical image segmentation tasks.**

**Overall Rating:** 7
**Confidence:** 3

**Other Quality Metrics:**

(a) Clarity of writing: good
(b) Clinical Significance: fair
(c) Methodological Novelty: good
(d) Experiments and Results: good

**Questions For The Authors:**

No question

**Strengths:**

The strength is the innovative integration of xLSTM with the U-Net architecture, which effectively combines the advantages of convolutional layers for local feature extraction and xLSTM for managing long-range dependencies. This hybrid approach addresses the limitations of traditional CNNs and ViTs and provides a more robust and efficient solution for biomedical image segmentation. The model consistently outperforms existing state-of-the-art methods across various datasets, including 2D and 3D medical images, demonstrating superior accuracy and efficiency.

**Summary Of The Paper:**

The paper introduces xLSTM-UNet, a novel image segmentation architecture that integrates Vision-LSTM (xLSTM) with the traditional U-Net structure. This model aims to address the limitations of Convolutional Neural Networks (CNNs) and Vision Transformers (ViTs) in handling long-range dependencies and computational overhead in biomedical image segmentation. xLSTM-UNet combines the local feature extraction capabilities of convolutional layers with the long-range dependency management of xLSTM. The proposed architecture outperforms existing CNN-based, Transformer-based, and Mamba-based models in various biomedical segmentation tasks, including abdominal MRI, endoscopy, and microscopy images. The study highlights xLSTM-UNet’s potential in advancing 2D and 3D medical image analysis and suggests its broader applicability across multiple domains. The paper’s experimental results and comprehensive validations demonstrate the model's superior performance and efficiency.

**Weaknesses:**

It lacks detailed discussion on potential limitations or drawbacks of the proposed model. While the paper demonstrates superior performance of xLSTM-UNet, it does not extensively address the increased computational complexity and resource demands that may accompany the integration of xLSTM into the U-Net architecture.

---

### Official Review · Reviewer_jyQk · 2024-08-09
**Review for submission 427**

**Overall Rating:** 6
**Confidence:** 4

**Other Quality Metrics:**

(a) Clarity of writing: good (b) Clinical Significance: fair (c) Methodological Novelty: fair (d) Experiments and Results: great

**Questions For The Authors:**

See weekness.

**Strengths:**

1. The paper is well-written and easy to follow.
2. The authors did extensive experiments on different image modality and both 2D and 3D segmentation task. The baseline method comparison is sufficient which including CNN, Transformer-based and mamba-based methods.

**Summary Of The Paper:**

The paper proposed xLSTM-UNet, which integrated xLSTM with UNet for medical image segmentation. Experiments on both 2D and 3D dataset shows improvement.

**Weaknesses:**

The model is a combinable of designed CNN decoder with Vision-LSTM encoder. There lacks of ablation study for decoder design.

Minor comments: it would be better to include some arrows to emphasize the difference in Fig.2.

---

### Official Review · Reviewer_x8PE · 2024-08-14
**xLSTM-UNet can be an Effective Backbone for 2D & 3D Biomedical Image Segmentation Better than its Mamba Counterparts**

**Overall Rating:** 8
**Confidence:** 5

**Other Quality Metrics:**

a) Clarity of writing;
good

(b) Clinical Significance;
good

 (c) Methodological Novelty;
good

 (d) Experiments and Results
good

**Questions For The Authors:**

Please address the weaknesses by answering:
- Any ablation studies?
- Compare the number of parameters in each method.
- Add speed comparisons.

**Strengths:**

- New architecture based on xLSTM-UNet.
- Demonstration on three different datasets.

**Summary Of The Paper:**

The paper introduces the xLST-Unet architecture for biomedical image segmentation. The method is successfully applied to three different biomedical datasets, and it has been found to give better results.

**Weaknesses:**

- There are no ablation studies.
- No comparisons of the number of parameters per method.
- No speed comparisons.

---

### Decision · Program_Chairs · 2024-09-23

Accept